evolution, behaviour, neuroscience

marmoset, vocal communication, vocal control, vocal learning, vocal plasticity

**Author for correspondence:**
Xiaoqin Wang
e-mail: xiaoqin.wang@jhu.edu

# Long-lasting vocal plasticity in adult marmoset monkeys

Lingyun Zhao[1], Bahar Boroumand Rad[2] and Xiaoqin Wang[1]

[1]Laboratory of Auditory Neurophysiology, Department of Biomedical Engineering, The Johns Hopkins University School of Medicine, Baltimore, MD 21205, USA
[2]Department of Biological Sciences, Towson University, Towson, MD 21252, USA

LZ, 0000-0003-4394-4063; XW, 0000-0002-3999-0772

Humans exhibit a high level of vocal plasticity in speech production, which allows us to acquire both native and foreign languages and dialects, and adapt to local accents in social communication. In comparison, non-human primates exhibit limited vocal plasticity, especially in adulthood, which would limit their ability to adapt to different social and environmental contexts in vocal communication. Here, we quantitatively examined the ability of adult common marmosets (*Callithrix jacchus*), a highly vocal New World primate species, to modulate their vocal production in social contexts. While recent studies have demonstrated vocal learning in developing marmosets, we know much less about the extent of vocal learning and plasticity in adult marmosets. We found, in the present study, that marmosets were able to adaptively modify the spectrotemporal structure of their vocalizations when they encountered interfering sounds. Our experiments showed that marmosets shifted the spectrum of their vocalizations away from the spectrum of the interfering sounds in order to avoid the overlap. More interestingly, we found that marmosets made predictive and long-lasting spectral shifts in their vocalizations after they had experienced a particular type of interfering sound. These observations provided evidence for directional control of the vocalization spectrum and long-term vocal plasticity by adult marmosets. The findings reported here have important implications for the ability of this New World primate species in voluntarily and adaptively controlling their vocal production in social communication.

## 1. Introduction

A hallmark of human vocal communication is voluntary vocal control and vocal learning throughout life [1]. This allows humans to adapt vocal production to suit communication needs. Vocal plasticity in humans has been demonstrated at different levels and time scales. Humans are able to manipulate many aspects of speech sounds in such situations as learning a foreign language or a local accent. These manipulations can be as simple as increasing the amplitude of voice when speaking in a noisy environment (e.g. the Lombard effect) [2] or as complicated as modifying spectrotemporal features of spoken words (e.g. compensatory changes in fundamental frequency [3] or vowel formant [4] when auditory feedback is altered; changes in formant frequency and spectral tilt in response to interfering noise [5–7]; modulations in phoneme structures in conversational contexts [8]). Such vocal modulations disappear when the noise or the interfering signal disappears and are considered short-term vocal plasticity. The most intriguing vocal plasticity is found when humans acquire novel vocal sounds with complex acoustic structures, for example, when learning one's native language during development [9] or learning a new language or dialect in adulthood [10,11]. This ability requires delicate control of vocal structures guided by auditory feedback and social contexts. Such long-lasting, persistent changes in vocal production (from days to years) [12] is considered long-term vocal plasticity.

Non-human primates, although evolutionarily close to humans, have been thought to have limited flexibility and plasticity in vocal production [13]. Particularly, monkey vocalizations are considered largely innate and not acquired through vocal production learning [14]. A large body of previous research has only found rudimentary levels of vocal plasticity in non-human primates. For example, the Lombard effect has been demonstrated in a number of monkey species, including macaques [15], marmosets [16] and tamarins [17]. Vocal modulations related to the Lombard effect included changes in amplitude [15–18], duration [16,17] or repetition [19,20]. In recent years, there has been an accumulation of evidence to indicate that non-human primates, in particular New World monkeys, may possess a higher level of vocal plasticity than previously thought [21]. It has been shown that marmosets are able to control the timing of vocal initiation in order to avoid interfering noises [22]. A recent study found that single phrases of marmoset phee calls can be interrupted by perturbation noises, indicating rapid control of vocal structures [23]. In developing marmosets, parental feedback was found to influence the maturation process of vocal behaviours [24–26]. In adult marmosets, it has been reported that modifications in spectro-temporal parameters of vocalizations occurred when there were changes in social contexts, such as when adding [27–30] or removing [31] individuals from an existing social group. Evidence of 'dialects' among geographically separated social groups has also been reported [32,33]. Some studies found that modifications in vocal structure occurred over a period of several weeks to several years under social [27,29,30] or environmental [34,35] influences, which suggested long-term plasticity [36]. While findings from field studies and behavioural observations have suggested a capacity of vocal learning and plasticity in adult marmosets, there is a lack of quantitative measurement to describe the degree of vocal plasticity and the extent of long-term plasticity in vocal production which is crucial to fully substantiate vocal plasticity and learning in adult marmosets.

These questions have motivated us to conduct well-controlled experiments in a laboratory condition where both short-term and long-term changes in the vocal structure can be quantitatively measured in adult marmosets that engage in vocal exchanges. In this study, we used interfering sounds with specific spectral contents to test the ability of adult marmosets in adaptively controlling the spectral parameters of their vocalizations. We found that when marmosets encountered interfering sounds with the spectra above or below the fundamental frequency of their vocalizations, they consistently shifted the fundamental frequency away from the spectra of the interfering sounds. Surprisingly, the shift in the fundamental frequency induced by a particular type of interfering sound persisted in the absence of interfering sounds up to several days after a test session, which suggested a long-term vocal plasticity. Our results provide further evidence for the voluntary control of spectrotemporal structures of vocalizations by adult marmosets and suggest specific types of external cues that may lead to context-related learning in the vocal behaviours of this species.

## 2. Material and methods

The subjects used in this study were four male adult common marmosets (Subject ID: 9606, 9001, 62U and 95Z)

housed in a captive colony at the Johns Hopkins University School of Medicine. The subjects were maintained on a diet consisting of a combination of monkey chow, fruit and yogurt, and had ad libitum access to water. All experimental procedures were approved by the Johns Hopkins University (JHU) Animal Care and Use Committee and in compliance with the guidelines of the National Institutes of Health (NIH).

### (a) Behavioural paradigm and apparatus

In each session, one subject was transported to a wire mesh recording cage (45 × 30 × 30 cm) inside a sound attenuating chamber [37] (figure 1*b*). Two loudspeakers (Cambridge Soundworks, M80, North Andover, MA, USA) placed 5 m apart were used to present sounds, including perturbation signals (speaker 1) and playback calls from the 'virtual conspecific' (speaker 2; see electronic supplementary material, method). The subject was engaged in an 'antiphonal calling' paradigm [38,39] and vocalized phee calls. The vocalizations were recorded through directional microphones (Sennheiser, ME66, Old Lyme, CT, USA) and saved to a computer. For each subject, we performed one experimental session each day. A typical session lasted 15–45 min, during which a subject would vocalize up to 90 calls. A custom-written MATLAB program detected the onset of the experimental subject's vocalizations and initiated the perturbation signal at a given delay at 50% probability. The detection of the vocalization was based on a combination of level (amplitude) threshold crossing and band-limited energy detection tuned to the typical marmoset vocalization frequency range (5–12 kHz). After a session ended, the subject was transported back to the colony to its home cage.

Previous studies in our laboratory had shown that when two marmosets were in isolation and were visually occluded from each other, one marmoset would produce phee calls spontaneously and exchange phee calls with the other marmoset, known as 'antiphonal calling' [38]. To engage a single marmoset (experimental subject) in antiphonal calling with experimental manipulations, we used a computer-simulated marmoset, the 'virtual conspecific'. The 'virtual conspecific' played back pre-recorded phee calls from another marmoset in our colony (presented by speaker 2). The playback calls were delivered in an interactive way, following the statistics of the natural antiphonal calling behaviour, so that the experimental subject made vocal exchanges in a similar way as it did with a real marmoset [39].

During the production of the first phrase of phees, the experimental subject received perturbation signals through speaker 1 (figure 1*b*). The perturbation signal was set to start at a delay after the onset of a call (about half of the median phrase length of the first phrase produced in baseline 1 sessions, see electronic supplementary material, table S1 'perturbation signal delay') with duration of one second (with a 100 ms linear ramp at the beginning to minimize transient effect). Two types of perturbation signal—high-frequency noise (HFN) and low-frequency noise (LFN)—were used, each of which was filtered from white noise such that it had only a high-frequency component or low-frequency component. The stop-band energy density was more than 60 dB lower than that in the pass-band. The cut-off frequency was selected to be about three standard deviations above or below the mean fundamental frequency of the first phrase in the baseline sessions (baseline 1 or baseline 2) at the time of perturbation signal delay, for HFN and LFN, respectively. The perturbation signals were presented at 75 dB SPL (Z-weighting, measured 0.8 m from speaker 1) and were calibrated for different cut-off frequencies and perturbation signal types.

The experiment started with the first group of baseline sessions (baseline 1; figure 1*c*), followed by a group of perturbation sessions (perturbation 1) with one type of perturbation signal (HFN or LFN). Then another group of baseline sessions (baseline 2) was tested, followed by a group of perturbation sessions

Proc. R. Soc. B 286: 20190817

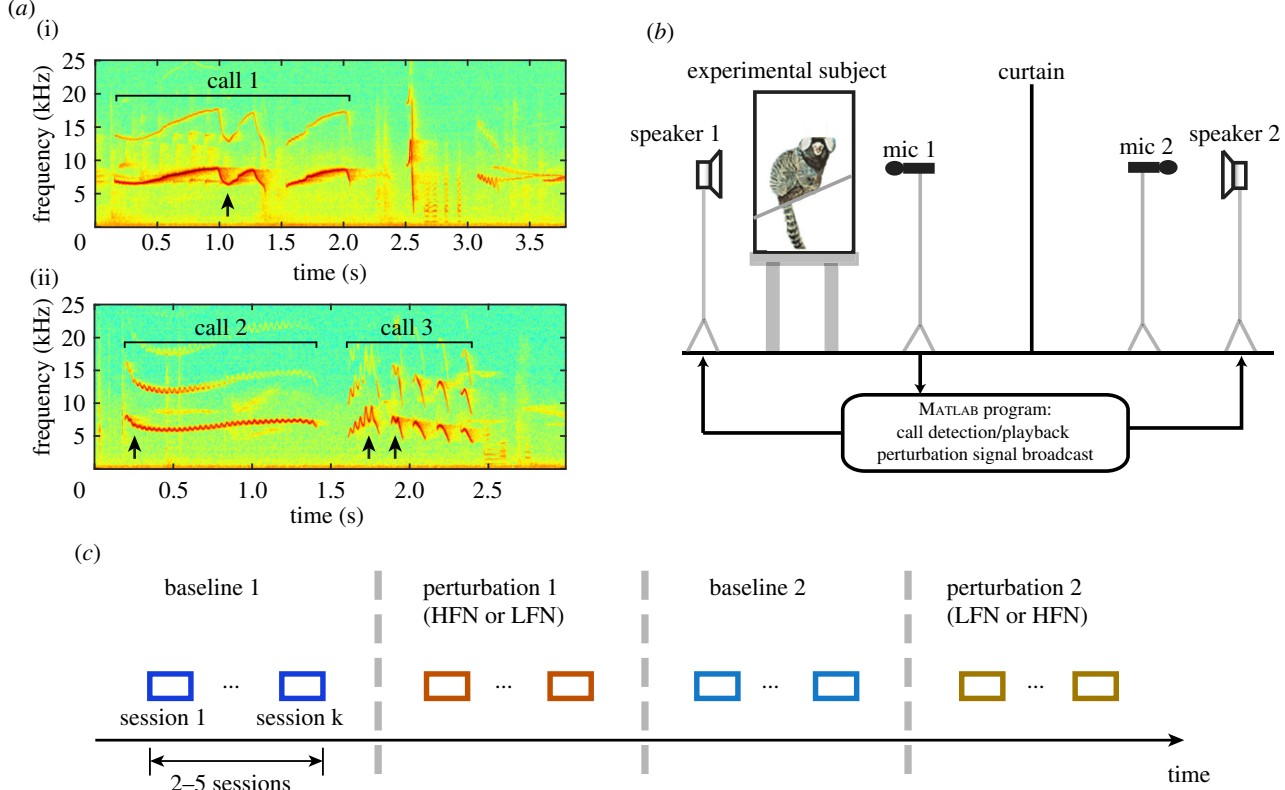

**Figure 1.** Natural variations of marmoset calls and the perturbation experiment set-up. (*a*) Variations in individual call structure in three example calls (indicated by brackets). Arrows indicate unusual modifications of call structure: a trillphee with two phrases seemingly concatenated (call 1, (i)); downward frequency modulation during a trill with prolonged duration (call 2, first arrow in (ii)); abrupt frequency modulation during trills (call 3, second and third arrows in (ii)). The *x*-axis is the time in these recorded clips. (*b*) Illustration of the set-up inside the recording chamber. The test cage for the experimental subject was located on the left. Vocalizations of the experimental subject were recorded by a microphone in front of the cage (mic 1) and were detected by a custom-written MATLAB program. A curtain is used to block the experimental subject's visual information about the other side of the room. (*c*) Illustration of experimental sessions (coloured blocks). There are four groups of sessions, as separated by the vertical dashed lines. HFN, high-frequency noise; LFN, low-frequency noise. The long arrow at the bottom indicates the progression time of the entire experiment for a single subject. (Online version in colour.)

**Table 1.** Perturbation signal types tested for each subject. HFN, high-frequency noise; LFN, low-frequency noise.

|  | 9606 | 62U | 9001 | 95Z |
|---|---|---|---|---|
| perturbation 1 | HFN | HFN | LFN | LFN |
| perturbation 2 | LFN | LFN | HFN | HFN |

(perturbation 2) for the other type of perturbation signal (LFN or HFN; table 1). Sessions were usually recorded on consecutive days, but in some cases, there were one or more non-recording days between adjacent sessions to increase the call rate during recording. Calls produced by the experimental subject during the perturbation sessions that received perturbation signals were labelled as 'perturbed' condition. Those in the same sessions but did not receive perturbation signals were labelled as 'not-perturbed' condition. Calls produced in the baseline sessions (either baseline 1 or baseline 2) were labelled as 'baseline' condition.

## (b) Data analysis

To analyse the effect of perturbation signals on the fundamental frequency, we used an analysis window in the latter half of the phee phrase, after the perturbation signal started (electronic supplementary material, table S1 and method). To capture the spontaneous changes of the fundamental frequency over time in the baseline sessions, the fundamental frequencies from baseline sessions were fitted by a piece-wise cubic spline function [40].

This is referred to as the 'fundamental frequency profile' of the baseline sessions (figure 2*b*(iii); electronic supplementary material, figure S1). Similar fitting was done to the perturbation sessions using not-perturbed and perturbed calls together to obtain the 'fundamental frequency profile' of perturbation sessions (figure 3*b*; electronic supplementary material, figure S1). To quantify the frequency shift on top of the spontaneous change, we calculated the relative frequency change (figures 2*b*(iii) and 3*c*; electronic supplementary material, figure S6), which is the difference between the fundamental frequency of each call in perturbation sessions (or in baseline sessions) and the value of the fundamental frequency profile of the directly preceding baseline sessions at the corresponding time point.

When comparing multiple groups of data, the Kruskal–Wallis test was applied and *post hoc* analysis with the Bonferroni correction was used to report significant difference for multiple comparisons. Significance was tested at an $\alpha$-level of 0.05.

## 3. Results

The common marmoset (*Callithrix jacchus*) is a highly vocal New World primate with a large vocal repertoire and rich vocal interactions among group members both in the wild and in captivity [41,42]. While marmosets have been shown to produce stereotypical species-specific vocalizations [41], they also produce vocalizations that are not described by the stereotypical call types, especially when they actively engage in vocal interactions with conspecifics in a rich social environment such as a large breeding colony (figure 1*a*).

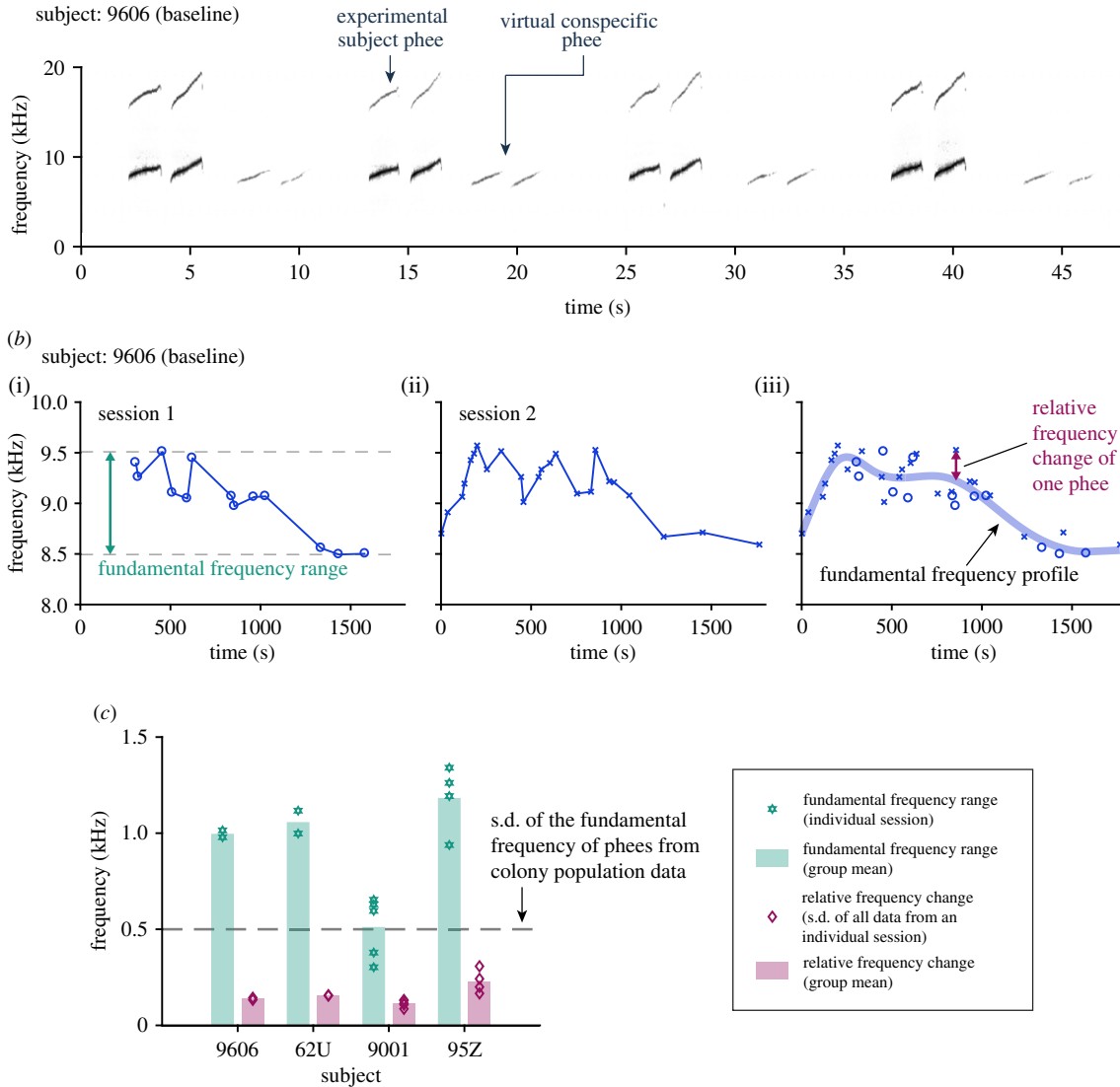

**Figure 2.** Spontaneous change in the fundamental frequency. (*a*) Example recording clip (spectrograms) from one of the baseline 1 sessions showing the exchange of phee calls from an experimental subject (9606) and the 'virtual conspecific' during antiphonal calling. (*b*) (i) Fundamental frequencies with respect to call onset time for the first session of baseline 1 from an experimental subject (9606). Each marker indicates a call. The *x*-axis is the time within individual sessions. Time zero indicates the start of a session. (ii) Same format as in (i), for the second session in baseline 1. (iii) The circles and crosses correspond to the data shown in (i) and (ii), respectively. The fundamental frequency profile (thick blue curve) is fitted from individual data points from all sessions in baseline 1 (see Material and methods, there are two sessions in total for this subject in baseline 1). The relative frequency change is indicated for one example phee call. (*c*) Comparison of two parameters (illustrated in *b*) of the spontaneous change in the fundamental frequency (in baseline 1) to the statistics of phee calls in the colony. Parameter 1 is the fundamental frequency range (see *b*(i)) (group mean: green bars; individual sessions: green stars). Parameter 2 is the standard deviation (s.d.) of relative frequency changes of all phee calls within an individual session (see *b*(iii)) (group mean: pink bars; s.d. of all data from an individual session: pink diamonds). The statistics of phee calls in the colony is the standard deviation of the fundamental frequency of phee calls from a population of marmosets recorded in the colony (dashed line) (see table IV of reference [41]). (Online version in colour.)

The example recordings in figure 1*a* showed two concatenated trillphees (i) and unusual spectral modulations of trill-like calls (ii). These less stereotyped vocalizations have large variations in spectrotemporal structures that are not observed when marmoset vocalizations are recorded in an impoverished social environment. Such large variations in call structures suggest that adult marmosets may possess greater flexibility in vocal production during vocal interactions than previously known. The natural environment that marmosets live contains interfering sounds from animal vocalizations and environmental noises, some of which have spectral energy distributed near or overlapping with the spectra of marmoset vocalizations [34,35]. For example, it was reported in a field study that low-frequency noises (2–8 kHz) were

presented in the environment due to anuran calls and insect-generated noise, whereas high-frequency noises (approx. 18 kHz) were found from the ambient sound that usually occurred in the afternoon [35]. Because marmosets' social behaviours depend on effective vocal communication in such an environment [43–45], we hypothesized that adult marmosets could learn about the acoustic environment and subsequently make predictive and long-lasting modifications of the spectrotemporal structure of their vocalizations to facilitate vocal communication with conspecifics. If marmosets were able to decipher the spectral contents of interfering sounds and modify their vocalizations to avoid spectral overlap, we would expect a directional shift in the spectra of their calls away from the interfering sounds.

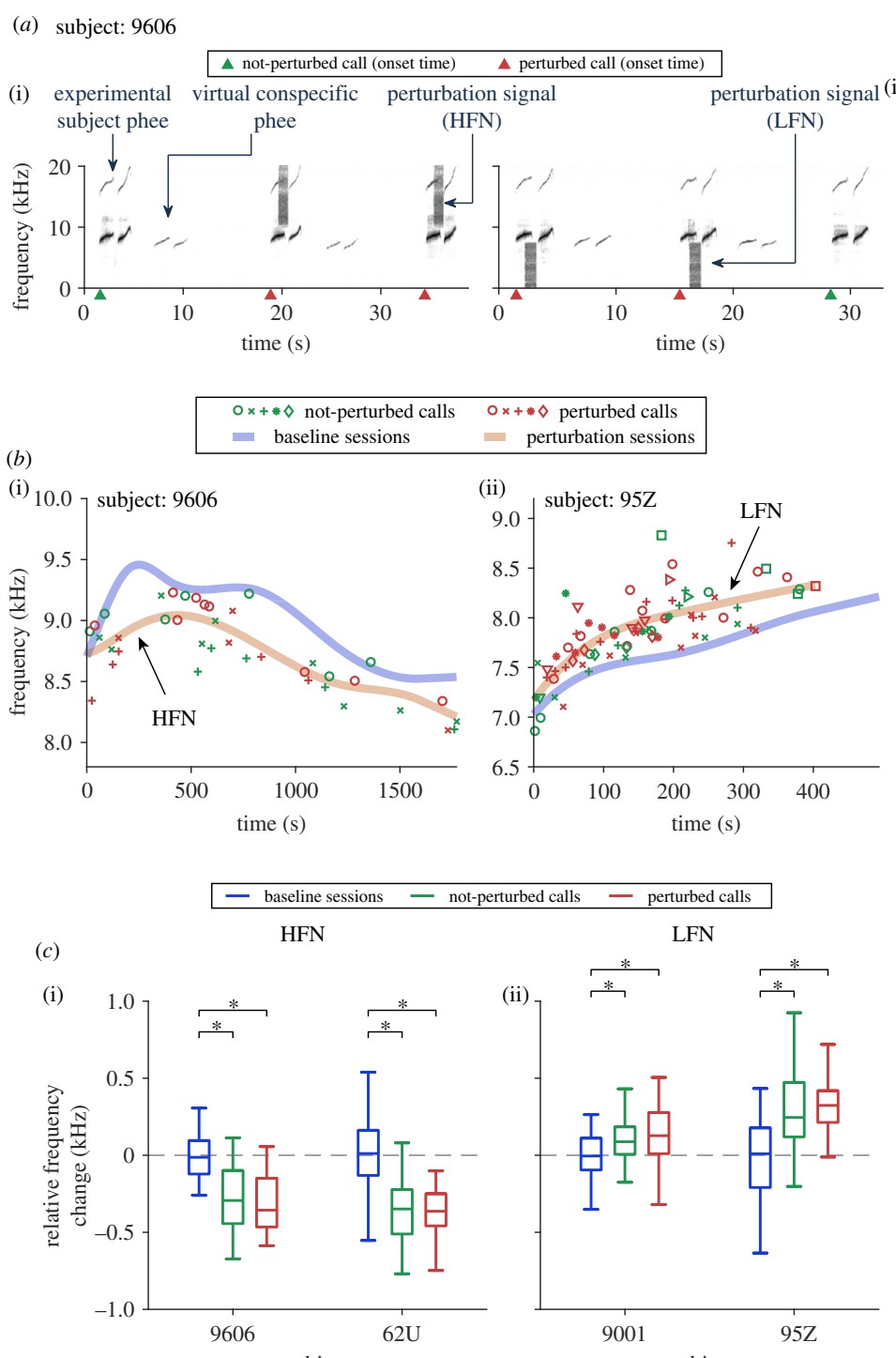

**Figure 3.** The fundamental frequency of phee calls shifts away from perturbation signal spectrum. (a) Example recordings from perturbation sessions. Perturbation signals overlapped with some of the phee calls from the experimental subject ((i) high-frequency noise, or HFN; (ii) low-frequency noise, or LFN). The green and red triangles mark the onset time of not-perturbed and perturbed calls, respectively. (b) Fundamental frequencies with respect to call onset time within perturbation 1 sessions compared to the fundamental frequency profile in baseline 1 sessions. Each marker indicates a call (same type of markers for calls in the same session)—green: not-perturbed; red: perturbed. The fundamental frequency profile (fitted from individual data points, figure 2b(iii) and Material and methods) of baseline 1 sessions is marked with a thick blue curve and that of perturbation 1 sessions is marked with a thick orange curve. The x-axis is the time within individual sessions. Time zero indicates the start of a session. (c) Statistical summary of the relative frequency change (Tukey boxplot) of calls in baseline (blue), not-perturbed (green) and perturbed (red) conditions for baseline 1 and perturbation 1 sessions. Medians: horizontal lines inside the boxes. First and third quartiles: lower and upper borders of the boxes. Inner fences: whiskers outside of the boxes. Outliers are not plotted. Any comparison showing a significant difference between conditions is indicated with asterisks. See electronic supplementary material, table S2 for the p-values. (Online version in colour.)

In order to test the above hypothesis, we used two types of interfering sounds to model some of the basic acoustic contexts that marmosets may encounter in their natural habitat. We presented these interfering sounds as perturbation signals to probe marmosets' ability to modify their vocalizations while

they made long duration phee calls (figure 1b). The phee call is a single or multi-phrase whistle-like call in which the majority of energy is centred at the fundamental frequency (figure 2a) [41,42,46]. Marmosets produce phee calls either spontaneously when isolated from others or when engaged in long-distance

vocal exchanges with conspecifics, known as 'antiphonal calling' [38]. In the following experiment, marmosets' vocal behaviours were evaluated in two types of experimental sessions conducted in a recording chamber outside of their colony. In 'perturbation sessions', perturbation signals were randomly delivered to 50% of phee calls vocalized by the experimental subject. In 'baseline sessions', no perturbation signals were presented. The baseline and perturbation sessions were interleaved (figure 1c): baseline 1, perturbation 1, baseline 2 and perturbation 2 (table 1). In all sessions, marmosets vocalized phee calls either spontaneously or evoked by the playback of pre-recorded phee calls in an 'antiphonal calling' paradigm [39] (figure 2a). The subject's vocalizations were recorded by a microphone, and the perturbation signals were delivered by an automated computer system shortly after the onset of a phee call (figure 1b; see Material and methods) [39]. The perturbation signals thus overlapped partially with the ongoing calls. We chose to present the perturbation signals in real-time with the calls in order to increase the likelihood marmosets modify their vocal structure. Previous studies have shown that if noise bursts were presented periodically in the background, marmosets would change the initiation time of vocal production, so that entire calls were shift in time to avoid being masked by noise [22]. After a session ended, the subject was transported back to the colony. In total, 1625 phee calls from four marmosets were recorded and included in the analysis.

## (a) Spontaneous change in fundamental frequency

Before we tested a subject's responses to perturbation signals, we first tracked the fundamental frequency of phee calls over the entire duration of a baseline session, which turned out to be a crucial analysis in revealing the effects of the perturbation. To our surprise, we observed that experimental subjects exhibited systematic changes in the fundamental frequency of phee calls during a baseline session. As the examples in figure 2b(i,ii) show, typically, the fundamental frequency of a subject's phee calls first increased, then slowly decreased, spanning a range of almost 1000 Hz. We refer to this trend as the fundamental frequency profile of baseline sessions (figure 2b(iii), thick blue curve; see Material and methods). This trend was observed in all baseline 1 sessions from each subject and in every marmoset tested in this experiment (electronic supplementary material, figure S1). This spontaneous but systematic change in the fundamental frequency of phee calls in the absence of any external stimuli is an interesting and unique property that has not been previously reported. It was not clear why marmosets displayed this systematic change over the course of a baseline session. Nevertheless, the characterization of this trend allowed us to reveal changes in their vocalizations in the presence of perturbation signals as described later.

To further quantify the spontaneous change in the fundamental frequency and validate the fundamental frequency profile, we calculated the range of the fundamental frequency of phee calls displayed by each experimental subject within a baseline session (figure 2c, green bars) and compared it to the standard deviation of the fundamental frequency of phee calls produced by marmosets in our colony (figure 2c, dashed line). For three out of four marmosets, the range of the fundamental frequency of phee calls was over 1000 Hz, twice the standard deviation of phee calls (approx. 500 Hz) recorded from a population of marmosets in our colony (22 animals,

12 841 phee calls, see tab. IV of reference [41]). The phee calls recorded in our marmoset colony were produced in a variety of contexts and presumably reflected the full extent of natural variations in spectrotemporal parameters of calls. Because the standard deviation of phee call fundamental frequency in our earlier study was calculated based on multiple marmosets, it was likely an overestimate of the range of the phee call fundamental frequency in an individual marmoset. In the light of these factors, the experimental subjects in the present study appeared to exhibit a surprisingly large range of variations in phee call fundamental frequency in baseline sessions. To further confirm that the changes in phee call fundamental frequency displayed by an experimental subject were not randomly distributed over the entire session, we calculated the relative frequency change of each phee call (i.e. the residue) by subtracting from its fundamental frequency the value of the fundamental frequency profile for that subject at the corresponding time point (figure 2b(iii)). Figure 2c shows that the relative frequency change in each subject was substantially smaller than the range of the fundamental frequency changes (figure 2c, pink bars versus green bars) and smaller than the standard deviation of the fundamental frequency of phee calls recorded in our colony (figure 2c, dashed line). These data indicate that the variations in the fundamental frequency of phee calls did not occur randomly, but followed a repeatable temporal pattern (characterized by the fundamental frequency profile; see further details in electronic supplementary material, text and figure S2). This phenomenon is interesting because it suggests that marmoset's vocal production system may have a greater capacity in voluntarily controlling spectral contents of its vocalizations than previously thought. Quantifying this systematic change in phee call fundamental frequency is a crucial step for analysing perturbation-induced frequency shifts.

## (b) Modulation of fundamental frequency induced by perturbation

After a subject's vocalizations were evaluated in a baseline session (baseline 1, figure 1c), they were then tested in a group of perturbation sessions (perturbation 1, figure 1c) in which one type of perturbation signals was delivered to approximately 50% of the vocalizations produced by the experimental subject. The perturbation signals were either high-frequency noise (HFN) or low-frequency noise (LFN) that were positioned above or below the fundamental frequency of the subject's phee calls (figure 3a). For perturbation 1 sessions, two subjects received HFN and the other two subjects received LFN (table 1). The same perturbation signal (HFN or LFN) was used in an experimental session.

Figure 3b shows two examples of perturbation 1 sessions, one subject was tested with HFN (figure 3b(i)) and the other subject was tested with LFN (figure 3b(ii)). Similar to the baseline sessions, we also observed a spontaneous change in the fundamental frequency of phee calls during perturbation sessions. Note that the fundamental frequency profile shifted downwards when the subject was tested with HFN, away from the spectrum of HFN (figure 3b(i): orange versus blue curve). By contrast, the fundamental frequency profile shifted upwards when the subject was tested with LFN, also away from the spectrum of LFN (figure 3b(ii): orange versus blue curve). Similar trends were observed in other subjects (electronic supplementary material, figure S1).

To quantify the frequency shifts in perturbation 1 sessions, we calculated the relative frequency change of each perturbed call (figure 3b, red symbols) with respect to the fundamental frequency profile of baseline 1 sessions at the corresponding time point. For the two subjects tested with HFN, the relative frequency change of perturbed calls was significantly lower than that of calls in baseline 1 sessions (figure 3c(i), red versus blue boxes), whereas the opposite was observed for the other two subjects tested with LFN (figure 3c(ii), red versus blue boxes; subject 9606: $\chi^2 = 33.2$, $p = 6.1 \times 10^{-8}$, subject 62U: $\chi^2 = 184.5$, $p = 8.8 \times 10^{-41}$, subject 9001: $\chi^2 = 14.4$, $p = 7.6 \times 10^{-4}$, subject 95Z: $\chi^2 = 29.6$, $p = 3.6 \times 10^{-7}$, the Kruskal–Wallis test, post hoc analysis with the Bonferroni corrections, $p < 0.05$ for each subject in each condition; see electronic supplementary material, table S2 for detailed $p$-values). In these perturbation sessions, only approximately 50% of calls received perturbation signals (see Material and methods). Interestingly, the fundamental frequency of not-perturbed calls (figure 3b, green symbols) in these perturbation sessions also showed similar trends of shifts as the perturbed calls (figure 3c, green versus red boxes). There was no significant difference in the magnitude of frequency shifts between perturbed and not-perturbed calls in either HFN or LFN perturbation sessions (figure 3c, $p > 0.05$ for each subject; see electronic supplementary material, table S2 for detailed $p$-values, the Kruskal–Wallis test, post hoc analysis with the Bonferroni corrections). This observation means that marmosets shifted the fundamental frequency of all phee calls within a perturbation session including those that were not directly perturbed. The observation that the fundamental frequency of not-perturbed phee calls also shifted away from perturbation signals in perturbation sessions suggests that marmosets in these experiments learned to maintain some degrees of memory of the context where a particular type of perturbation signals was expected, and they voluntarily modified the spectral characteristics of their vocalizations based on the memory of the context. Because the same type of perturbation signals (HFN or LFN) was used within an entire perturbation session, marmosets may have anticipated not only the occurrence of the perturbation signals but also their spectral contents. As a result, they strategically made predictive changes to all subsequent calls soon after a session started. This is also supported by the fact that larger frequency shifts usually occurred towards the later stage of the perturbation sessions (electronic supplementary material, figure S3).

## (c) Long-lasting effect of perturbation signals

Given the observation that marmosets made predictive changes to all calls within a perturbation session, we wondered whether this effect persisted beyond the end of perturbation sessions. After perturbation 1 sessions ended, we evaluated baseline vocalizations again (referred to as baseline 2 sessions, figure 1c) before testing the same animal with another set of perturbation sessions (referred to as perturbation 2 sessions, figure 1c). Figure 4a shows the data obtained from one marmoset (subject 9606). As expected, the fundamental frequency profile of the perturbation 1 sessions with HFN dropped below that of the baseline 1 sessions preceding it (figure 4a(i)). To our surprise, the fundamental frequency profile of the baseline 2 sessions following the perturbation 1 sessions did not return to that of the baseline 1 sessions, but instead it remained close to the fundamental frequency

profile of the perturbation 1 sessions (figure 4a(ii)). In other words, the fundamental frequency profile of the baseline 2 sessions shifted in the same direction (become lower in frequency) as the fundamental frequency profile of the perturbation 1 sessions. Also as expected, the fundamental frequency profile of the perturbation 2 sessions with LFN rose higher in frequency than that of the baseline 2 sessions preceding it (figure 4a(iii)).

We performed the above analyses in three of the four marmosets in this study (see electronic supplementary material, figure S3 for the number of sessions and days for each subject). Because the fourth marmoset (subject 9001) was used for other experiments after perturbation 1, baseline 2 data were not available in this subject (see electronic supplementary material, figure S3). Figure 4b compared the shift in fundamental frequency relative to that of baseline 1 sessions in perturbation 1, baseline 2 and perturbation 2 sessions. It is clear in all three subjects that the fundamental frequency of the baseline 2 sessions shifted towards the fundamental frequency of the preceding perturbation 1 session (figure 4b). In fact, there was no significant difference in median fundamental frequency between baseline 2 and perturbation 1 sessions in any of the three marmosets tested ($p > 0.05$; see electronic supplementary material, table S3 for detailed $p$-values, the Kruskal–Wallis test, post hoc analysis with the Bonferroni corrections; Kruskal–Wallis test results—subject 9606: $\chi^2 = 52.7$, $p = 2.2 \times 10^{-11}$, subject 62U: $\chi^2 = 189.3$, $p = 8.7 \times 10^{-41}$, subject 95Z: $\chi^2 = 48.4$, $p = 1.7 \times 10^{-10}$; figure 4b). The fundamental frequency of the HFN or LFN perturbation sessions (either as perturbation 1 or perturbation 2), however, was significantly different from the fundamental frequency of the baseline session preceding it ($p < 0.05$; see electronic supplementary material, table S3 for detailed $p$-values, the Kruskal–Wallis test, post hoc analysis with the Bonferroni corrections; figure 4b).

In short, for all three tested subjects, the fundamental frequency of phee calls shifted in perturbation 1 sessions (downwards or upwards depending on perturbation signal types being HFN or LFN) and then stayed at the shifted values in baseline 2 sessions. The fundamental frequency shifted again in perturbation 2 sessions (downwards or upwards depending on perturbation signal types being HFN or LFN; figure 4b; electronic supplementary material, figures S5 and S6, and table S4). For the fourth marmoset (subject 9001), the fundamental frequency shifted predictably in both perturbation 1 and perturbation 2 sessions compared with preceding baseline sessions, respectively (electronic supplementary material, figures S4–S6). Therefore, all four marmosets showed upward or downward shifts in fundamental frequency depending on the spectra of perturbation signals. These results provide intriguing evidence suggesting that marmosets may have learnt and remembered the context in which a particular type of perturbation signal was delivered. To minimize the interference to their vocal exchanges in an antiphonal calling setting, marmosets produced phee calls with frequency shifts towards a predictive direction in anticipation of the perturbation signal that they had recently experienced. Because experimental sessions were usually separated by 1 day (sometimes multiple days, electronic supplementary material, figure S3), their memory of the context lasted longer than at least 1 day. This long-lasting effect provides further evidence to suggest that marmosets are able to voluntarily control the production of the spectrotemporal structure of their vocalizations and use this ability to guide vocal communications in different social contexts.

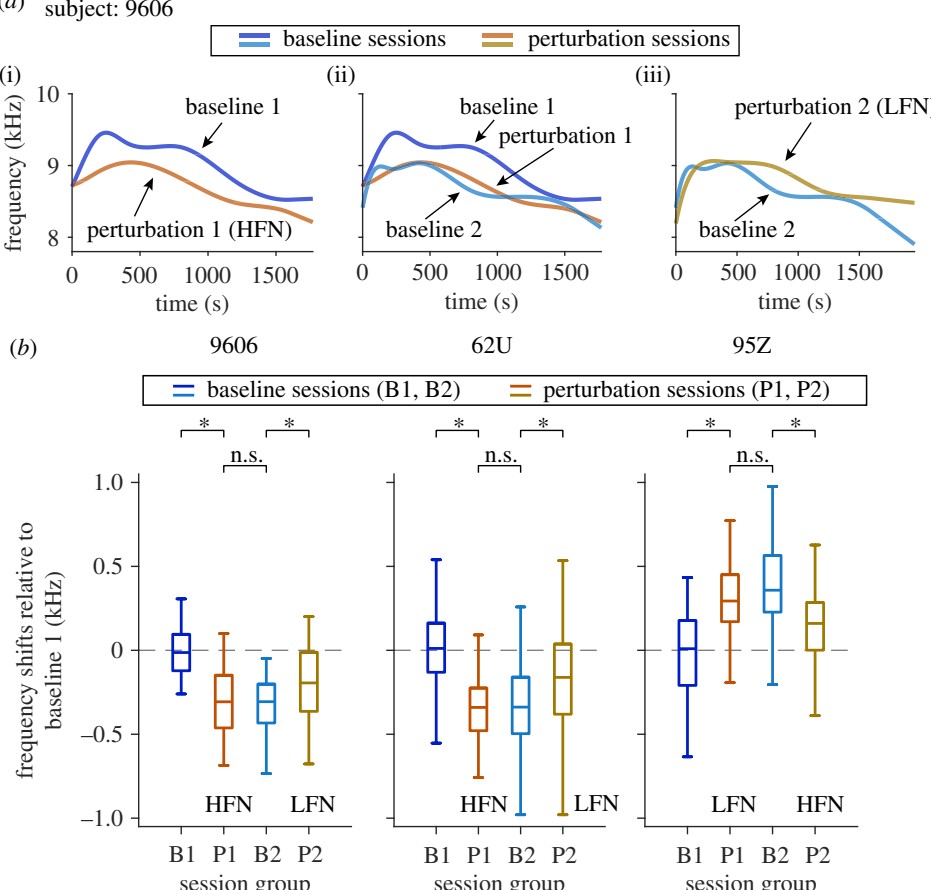

**Figure 4.** Long-lasting effects of the perturbation signals. (*a*) Change in fundamental frequency profile between adjacent session groups of an example subject (9606). Blue and cyan: baseline sessions. Orange and brown: perturbation sessions. The *x*-axis is the time within individual sessions. Time zero indicates the start of a session. (*b*) Fundamental frequency shifts (Tukey boxplot) over time in experimental session order, with respect to baseline 1 sessions (for subjects 9606, 62U and 95Z). Medians: horizontal lines inside the boxes. First and third quartiles: lower and upper borders of the boxes. Inner fences: whiskers outside of the boxes. Outliers are not plotted. Calls in perturbation sessions include both the not-perturbed and the perturbed calls. B1: baseline 1; P1: perturbation 1; B2: baseline 2; P2: perturbation 2. Asterisks indicate significant differences. n.s., not significant. See electronic supplementary material, table S3 for the *p*-values. (Online version in colour.)

## 4. Discussion

The present study provided three important observations. First, adult marmosets exhibited considerable variations in their vocalizations in colony or testing environments (figures 1*a* and 2), suggesting that these animals have the ability to produce a broad range of spectral and temporal parameters. Second, adult marmosets shifted their vocalization spectrum away from the spectrum of interfering sounds when they encountered or, more interestingly, anticipated the interfering sounds (figure 3), which suggests the voluntary directional control of the spectrotemporal structure of vocalizations. Third, and most importantly, the spectral shifts in vocalizations initially induced by perturbation signals lasted many days after perturbation sessions ended and occurred in the absence of the perturbation signals (figure 4), which suggests long-term plasticity in the marmoset's vocal production system. Together, these results produced further evidence of long-term vocal plasticity in marmosets' vocal production, which can potentially benefit vocal communications in their natural habitats. A limitation of the present study is the limited experimental parameters tested due to the challenging nature of these experiments. Future studies shall evaluate the above conclusions in a wider range of acoustic contexts.

Previous studies in non-human primates had described gross changes in vocal parameters such as amplitude and duration (e.g. Lombard effect) when perturbation signals were presented [16,18], which have largely been attributed to factors other than cognitive functions [47,48]. One recent study has shown evidence for spectral adjustment in tamarin vocalizations with noise perturbation [49]. However, it did not dissociate the changes from the Lombard effect. Studies in birds [50–52] and humans [6,53] also showed similar spectral changes secondary to amplitude changes when tested with noise perturbation. The present study provided compelling evidence for context-dependent, directional control of the vocal structure in marmosets (figure 3; electronic supplementary material, figure S6, discussion).

The experiments reported here provided two crucial observations to further substantiate the extent of long-term plasticity in vocal production by adult marmosets. First, the fundamental frequency of not-perturbed phee calls shifted away from perturbation signals in perturbation sessions (figure 3*c*; electronic supplementary material, figure S6), which indicated that the modification of the vocal structure persisted beyond the perturbation signals. It is possible that the marmosets in these experiments learned and memorized the context where a particular type of perturbation signals (HFN or LFN) would be expected and voluntarily modified the spectral characteristics of their vocalizations based on the memory of that context. Since the time interval between not-perturbed calls and the preceding perturbed calls ranged from several seconds to minutes (figure 3*a*; electronic supplementary material, figure S1), the memory of the context lasted for at

least that long. Second, the fundamental frequency of phee calls in baseline 2 sessions did not return to that of initial baseline values measured in baseline 1 sessions; instead, it stayed close to that of preceding perturbation 1 sessions (figure 4). In other words, the marmosets continued to anticipate the perturbation signals that they had recently experienced and produced phee calls with frequency shifts in the same direction as in the previous perturbation sessions even though they were not being perturbed in the baseline 2 sessions. This unexpected long-lasting spectral shift in vocalizations suggests that the anticipation of a particular type of perturbation signals took place beyond perturbation sessions. Therefore, the marmoset's memory of which type of perturbation signals was delivered appeared to last longer than 1 day because test sessions were usually separated by one to several days (electronic supplementary material, figure S3). This is the first time such evidence has been revealed in adult non-human primates including marmosets. This effect is also interesting because the marmosets in this study stayed in a different environment between testing sessions. They experienced completely different acoustic and social context in the colony room between testing sessions, but still exhibited frequency shift when they were brought back to the recording chamber for testing, which indicated some level of vocal memory of the context of the testing sessions.

The long-lasting effect discussed above provided clear evidence on the marmoset's ability to voluntarily control the production of the spectrotemporal structure of their vocalizations in the absence of perturbation signals. It suggests that marmosets use auditory information to guide long-term plasticity in vocal production. It also suggests the possibility of context-based vocal learning by adult marmosets. This finding also shows the importance of measuring the fundamental frequency in the baseline condition again before perturbation 2 sessions in our experimental design. Had we compared the fundamental frequency of calls in perturbation 2 sessions to those in baseline 1 sessions, we would not have been able to reveal the frequency shift in the perturbation 2 sessions (figure 4b; electronic supplementary material, figure S4).

What we observed in the present study appears to be a higher level of vocal control by a non-human primate species than what has been shown in previous studies. If marmosets simply showed the Lombard effect, then the change in the fundamental frequency of their phee calls would be linked to the amplitude change as predicted from previous studies. In this case, we should expect the fundamental frequency to shift upwards for either type of perturbation noise (HFN or LFN). However, our results showed a consistent upward or downward shift in fundamental frequency depending on the spectral property of the perturbation signal (figure 3; electronic supplementary material, figure S6). To our knowledge, this is the first study in non-human primates that demonstrates directional spectral shifts of vocalizations which suggests that marmosets have the ability to systematically modify spectrotemporal structures of their vocalizations guided by external acoustic cues.

In our experimental setting, marmosets maintained antiphonal calling with a 'virtual conspecific' either in a quiet environment or in the presence of perturbation noises. The frequency shifts in phee calls likely helped marmosets to minimize the noise interference to their vocal exchanges. In the natural habitat, vocal communication is known to play an important role in marmosets' social behaviours [43,44]. Their vocalizations are prone to various types of noise interference [35], imposing challenges for marmosets' social interactions in a natural environment. The ability to voluntarily control their vocal production and to learn and memorize acoustic, behavioural and social contexts can help marmosets guide vocal communications in a natural environment.

**Ethics.** All research was performed in accordance with NIH guidelines. These experiments were approved by the JHU Animal Care and Use Committee.

**Data accessibility.** The dataset has been made available in the Dryad repository at https://doi.org/10.5061/dryad.7nq1c6s [54].

**Authors' contributions.** L.Z. and X.W. designed the study; L.Z. and B.B.R. collected data and performed analyses; L.Z. and X.W. wrote the paper.

**Competing interests.** We declare that we have no competing interests.

**Funding.** This work was supported by NIH grant nos. DC005808, DC014503.

**Acknowledgement.** We thank Nate Sotuyo and Shanequa Smith for assistance with animal care. We thank Reza Shadmehr for helpful discussions about the data. We thank Cynthia Moss, Jinhong Luo, Michael Osmanski, Scott Sterrett and Xindong Song for comments on the manuscript. We thank Joon Lee, Nicolas Gutierrez for help with experiments and data analysis.

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
