## [Reviewer comments · Proceedings of the Royal Society B: Biological Sciences]

Review History

RSPB-2019-0205.R0 (Original submission)

Review form: Reviewer 1

Recommendation

Accept with minor revision (please list in comments)

Scientific importance: Is the manuscript an original and important contribution to its field?

Good

General interest: Is the paper of sufficient general interest?

Excellent

Quality of the paper: Is the overall quality of the paper suitable?

Good

Is the length of the paper justified?

Yes

Should the paper be seen by a specialist statistical reviewer?

No

Do you have any concerns about statistical analyses in this paper? If so, please specify them explicitly in your report.

No

It is a condition of publication that authors make their supporting data, code and materials available - either as supplementary material or hosted in an external repository. Please rate, if applicable, the supporting data on the following criteria.

Is it accessible?

No

Is it clear?

N/A

Is it adequate?

N/A

Do you have any ethical concerns with this paper?

No

Comments to the Author

One issue which mars this ms is a number of overstatements. Here for the Abstract:

"Non-human primates, however, have long been thought to have little plasticity in vocal production, especially in adulthood, which would limit their ability in carrying out effective vocal communication in natural environment". Given the authors note that developing marmosets can show vocal plasticity, and the Introduction reviews many articles (refs 17-19, 23-32) showing adult plasticity, the first part is an overstatement and the second part is not logical since the absence of vocal plasticity in other species is not correlated with an inability to carry out effective communication which must be viewed in any evolutionary sense as environmental niche-appropriate communication.

What distinguishes this study is NOT a novel demonstration of adult vocal plasticity in marmosets, but the precise quantification (c.f lines 78-80) of such plasticity under their lab-based test and observational conditions. In this context, the claim that the ability of adult marmosets to systematically alter vocalizations spectrotemporal structures during vocal interactions with conspecifics in a manner dependent on acoustic context and experience is crucial to demonstrate vocal plasticity and learning, is again exaggerated - it is a strong and very important manifestation of vocal plasticity and learning but the basic demonstration of a capacity for vocal plasticity and learning is already present in the studies reviewed.

Thus the claim at the end of the Introduction that "Our results provide new evidence for the voluntary control of spectrotemporal structures of vocalizations by adult marmosets and suggest the possibility of context related learning in the vocal behaviors of this species." is exaggerated. This study provides further evidence for the voluntary control of spectrotemporal structures of vocalizations by adult marmosets and is consistent with evidence for context related learning in the vocal behaviors of this species. This doesn't devalue this study which is a very well controlled exercise that gives confidence in such vocal learning but it does mean the authors must mute the overstatements.

Minor errors:

English expression can be improved. Here are some early examples: line 27 environments; l29

contexts; l30 of instead of on; similarly in the Discussion (note this is in the context that this was a well-written paper otherwise)

Fig 1A, the call shown in the spectrograms must be identified in the legend so the figure is stand-alone.

Review form: Reviewer 2

Recommendation

Major revision is needed (please make suggestions in comments)

Scientific importance: Is the manuscript an original and important contribution to its field?

Excellent

General interest: Is the paper of sufficient general interest?

Good

Quality of the paper: Is the overall quality of the paper suitable?

Good

Is the length of the paper justified?

Yes

Should the paper be seen by a specialist statistical reviewer?

Yes

Do you have any concerns about statistical analyses in this paper? If so, please specify them explicitly in your report.

No

It is a condition of publication that authors make their supporting data, code and materials available - either as supplementary material or hosted in an external repository. Please rate, if applicable, the supporting data on the following criteria.

Is it accessible?

N/A

Is it clear?

N/A

Is it adequate?

N/A

Do you have any ethical concerns with this paper?

No

Comments to the Author

A fascinating and well-written read! However, authors should address the following before publication.

1. Describe and include any research on humans that parallels specifically the interfering talker

Marmoset scenario. The Lombard effect and other effects included are not adequate.

2. Define "antiphonal calling" for the reader.
3. Be more specific in defining low and high frequency noise --- does this mimic an interfering call? If not, then what? What specific environmental noises are also being simulated. How accurate are the simulated sounds in reproducing interfering calls and true environmental noises?
4. The perturbation signal occurs "after the onset of a phee call". Is it overlapping the call and to what extent? Would there be a different effect if not overlapping?
5. Please describe how the virtual calls and the perturbations are actually implemented with the Marmoset calls in real time. The authors say nothing about this!
6. The statistical analysis appears careful but needs further clarity.
7. In a real dialogue, would a receiving marmoset not get confused if the caller is anticipating a shift not present and thus would the learning effect stop soon?
8. In the discussion, authors should tone down the generalizations based on the limited experimental setup.
9. Figures 1,2 and parts of 3,4 are very confusing in terms of timing of the various signal components. Please clarify.

4.

Decision letter (RSPB-2019-0205.R0)

06-Mar-2019

Dear Dr Wang:

I am writing to inform you that your manuscript RSPB-2019-0205 entitled "Long-lasting Vocal Plasticity in Adult Marmoset Monkeys" has, in its current form, been rejected for publication in Proceedings B.

This action has been taken on the advice of referees, who have recommended that substantial revisions are necessary. With this in mind we would be happy to consider a resubmission, provided the comments of the referees are fully addressed. However please note that this is not a provisional acceptance. To add to the referees' comments, please discuss the social experience that your subjects had and consider how these results may reflect their behavior in a more natural context, and carefully reframe the manuscript to ensure that you do not overstate your findings.

Sincerely,

Prof Sarah F. Brosnan
Editor, Proceedings B
mailto: proceedingsb@royalsociety.org

Associate Editor

Comments to Author:

Your manuscript has now been evaluated by two experts in the field. Although they both found a lot to like about your study, they have raised some concerns. I am therefore recommending rejection of the manuscript in its present form. Having said that, I would also encourage you to re-submit a revised manuscript. In your revision, it would be useful if you could make sure that the statistical analyses are made as clear and as accessible as possible.

Reviewer(s)' Comments to Author:

Referee: 1

Comments to the Author(s)

One issue which mars this ms is a number of overstatements. Here for the Abstract: "Non-human primates, however, have long been thought to have little plasticity in vocal production, especially in adulthood, which would limit their ability in carrying out effective vocal communication in natural environment". Given the authors note that developing marmosets can show vocal plasticity, and the Introduction reviews many articles (refs 17-19, 23-32) showing adult plasticity, the first part is an overstatement and the second part is not logical since the absence of vocal plasticity in other species is not correlated with an inability to carry out effective communication which must be viewed in any evolutionary sense as environmental niche-appropriate communication.

What distinguishes this study is NOT a novel demonstration of adult vocal plasticity in marmosets, but the precise quantification (c.f lines 78-80) of such plasticity under their lab-based test and observational conditions. In this context, the claim that the ability of adult marmosets to systematically alter vocalizations spectrottemporal structures during vocal interactions with conspecifics in a manner dependent on acoustic context and experience is crucial to demonstrate vocal plasticity and learning, is again exaggerated - it is a strong and very important manifestation of vocal plasticity and learning but the basic demonstration of a capacity for vocal plasticity and learning is already present in the studies reviewed.

Thus the claim at the end of the Introduction that "Our results provide new evidence for the voluntary control of spectrottemporal structures of vocalizations by adult marmosets and suggest the possibility of context related learning in the vocal behaviors of this species." is exaggerated. This study provides further evidence for the voluntary control of spectrottemporal structures of vocalizations by adult marmosets and is consistent with evidence for context related learning in the vocal behaviors of this species. This doesn't devalue this study which is a very well controlled

exercise that gives confidence in such vocal learning but it does mean the authors must mute the overstatements.

Minor errors:

English expression can be improved. Here are some early examples: line 27 environments; l29 contexts; l30 of instead of on; similarly in the Discussion (note this is in the context that this was a well-written paper otherwise)

Fig 1A, the call shown in the spectrograms must be identified in the legend so the figure is stand-alone.

Referee: 2

Comments to the Author(s)

A fascinating and well-written read! However, authors should address the following before publication.

1. Describe and include any research on humans that parallels specifically the interfering talker Marmoset scenario. The Lombard effect and other effects included are not adequate.
 2. Define "antiphonal calling" for the reader.
 3. Be more specific in defining low and high frequency noise --- does this mimic an interfering call? If not, then what? What specific environmental noises are also being simulated. How accurate are the simulated sounds in reproducing interfering calls and true environmental noises?
 4. The perturbation signal occurs "after the onset of a phee call". Is it overlapping the call and to what extent? Would there be a different effect if not overlapping?
 5. Please describe how the virtual calls and the perturbations are actually implemented with the Marmoset calls in real time. The authors say nothing about this!
 6. The statistical analysis appears careful but needs further clarity.
 7. In a real dialogue, would a receiving marmoset not get confused if the caller is anticipating a shift not present and thus would the learning effect stop soon?
 8. In the discussion, authors should tone down the generalizations based on the limited experimental setup.
 9. Figures 1,2 and parts of 3,4 are very confusing in terms of timing of the various signal components. Please clarify.
- 4.

Author's Response to Decision Letter for (RSPB-2019-0205.R0)

See Appendix A.

RSPB-2019-0817.R0

Review form: Reviewer 2

Recommendation

Accept as is

Scientific importance: Is the manuscript an original and important contribution to its field?

Good

General interest: Is the paper of sufficient general interest?

Acceptable

Quality of the paper: Is the overall quality of the paper suitable?

Good

Is the length of the paper justified?

Yes

Should the paper be seen by a specialist statistical reviewer?

No

Do you have any concerns about statistical analyses in this paper? If so, please specify them explicitly in your report.

No

It is a condition of publication that authors make their supporting data, code and materials available - either as supplementary material or hosted in an external repository. Please rate, if applicable, the supporting data on the following criteria.

Is it accessible?

No

Is it clear?

N/A

Is it adequate?

N/A

Do you have any ethical concerns with this paper?

No

Comments to the Author

Authors have satisfactorily addressed reviewer concerns.

Decision letter (RSPB-2019-0817.R0)

28-May-2019

Dear Dr wang

I am pleased to inform you that your manuscript RSPB-2019-0817 entitled "Long-lasting Vocal Plasticity in Adult Marmoset Monkeys" has been accepted for publication in Proceedings B.

The referee(s) have recommended publication, but also suggest some minor revisions to your manuscript. Therefore, I invite you to respond to the referee(s)' comments and revise your manuscript. Because the schedule for publication is very tight, it is a condition of publication that you submit the revised version of your manuscript within 7 days. If you do not think you will be able to meet this date please let us know.

Sincerely,

Prof Sarah F Brosnan
Editor, Proceedings B
mailto: proceedingsb@royalsociety.org

Associate Editor
Board Member
Comments to Author:

I am pleased to recommend acceptance of your fascinating manuscript. Congratulations.

Reviewer(s)' Comments to Author:

Referee: 2

Comments to the Author(s).
Authors have satisfactorily addressed reviewer concerns.

Decision letter (RSPB-2019-0817.R1)

04-Jun-2019

Dear Dr wang

I am pleased to inform you that your manuscript entitled "Long-lasting Vocal Plasticity in Adult Marmoset Monkeys" has been accepted for publication in Proceedings B.

Open Access

Paper charges

Sincerely,

Appendix A

Responses to Referees' Comments (RSPB-2019-0205)

We thank the two reviewers for their constructive and helpful comments. In the revised manuscript, we have carefully addressed all concerns raised by the reviewers. In addition to the specific changes in response to reviewers' comments, we have added more detailed clarifications in the figure legend. We hope the reviewers find our responses satisfactory and we'd be happy to address any further questions.

The following is a point-by-point list of all changes made in response to the comments by the two reviewers. The page and line numbers refer to those of the revised manuscript (please see the document marked with "tracked changes"). The reviewers' comments are quoted and marked by blue font, followed by our response in black font.

Associate Editor

Comments to Author:

Your manuscript has now been evaluated by two experts in the field. Although they both found a lot to like about your study, they have raised some concerns. I am therefore recommending rejection of the manuscript in its present form. Having said that, I would also encourage you to re-submit a revised manuscript. In your revision, it would be useful if you could make sure that the statistical analyses are made as clear and as accessible as possible.

We have expanded our report for statistical analysis in the main text according to the suggestion by Reviewer 2.

Reviewer(s)' Comments to Author:

Referee: 1

Comments to the Author(s)

One issue which mars this ms is a number of overstatements. Here for the Abstract: "Non-human primates, however, have long been thought to have little plasticity in vocal production, especially in adulthood, which would limit their ability in carrying out effective vocal communication in natural environment". Given the authors note that developing marmosets can show vocal plasticity, and the Introduction reviews many articles (refs 17-19, 23-32) showing adult plasticity, the first part is an overstatement and the second part is not logical since the absence of vocal plasticity in other species is not correlated with an inability to carry out effective communication which must be viewed in any evolutionary sense as environmental niche-appropriate communication.

We appreciate the reviewer's recognition of the importance of the results presented in this manuscript and understand the reviewer's concerns of overstatements. In light of the reviewer's critiques, we have carefully re-worded a number of statements throughout the manuscript to

make sure we do not overstate the findings of this study. The above-mentioned statement in the abstract has been revised as follows (line 24-28).

“Humans exhibit a high level of vocal plasticity in speech production which allows us to acquire both native and foreign languages, dialects and adapt to local accents in social communication. In comparison, non-human primates exhibit limited vocal plasticity, especially in adulthood, which would limit their ability to adapt to different social and environmental contexts in vocal communication.”

What distinguishes this study is NOT a novel demonstration of adult vocal plasticity in marmosets, but the precise quantification (c.f lines 78-80) of such plasticity under their lab-based test and observational conditions. In this context, the claim that the ability of adult marmosets to systematically alter vocalizations spectrotemporal structures during vocal interactions with conspecifics in a manner dependent on acoustic context and experience is crucial to demonstrate vocal plasticity and learning, is again exaggerated - it is a strong and very important manifestation of vocal plasticity and learning but the basic demonstration of a capacity for vocal plasticity and learning is already present in the studies reviewed.

We thank the reviewer for your comment! We have revised the above-mentioned statement in the introduction as follows (line 96-99).

“While findings from field studies and behavioral observations have suggested a capacity of vocal learning and plasticity in adult marmosets, there is a lack of quantitative measurement to describe the degree of vocal plasticity and the extent of long-term plasticity in vocal production which is crucial to fully substantiate vocal plasticity and learning in adult marmosets.”

Thus the claim at the end of the Introduction that "Our results provide new evidence for the voluntary control of spectrotemporal structures of vocalizations by adult marmosets and suggest the possibility of context related learning in the vocal behaviors of this species." is exaggerated. This study provides further evidence for the voluntary control of spectrotemporal structures of vocalizations by adult marmosets and is consistent with evidence for context related learning in the vocal behaviors of this species. This doesn't devalue this study which is a very well controlled exercise that gives confidence in such vocal learning but it does mean the authors must mute the overstatements.

We have revised the above-mentioned statement as follows (line 120-123).

“Our results provide further evidence for the voluntary control of spectrotemporal structures of vocalizations by adult marmosets and suggest specific types of external cues that may lead to context related learning in the vocal behaviors of this species.”

Minor errors:

English expression can be improved. Here are some early examples: line 27 environments; 129 contexts; 130 of instead of on; similarly in the Discussion (note this is in the context that this was a well-written paper otherwise)

Thanks for pointing out these errors! We have corrected these errors and carefully checked the entire manuscript again.

Fig 1A, the call shown in the spectrograms must be identified in the legend so the figure is stand-alone.

We have labeled the example calls in Fig 1A and added detailed explanations in the figure legend.

Referee: 2

Comments to the Author(s)

A fascinating and well-written read! However, authors should address the following before publication.

1. Describe and include any research on humans that parallels specifically the interfering talker Marmoset scenario. The Lombard effect and other effects included are not adequate.

We've added this in the first paragraph of introduction (line 67-69). We've also added a paragraph in discussion (paragraph 2) to compare our results to previous studies.

2. Define "antiphonal calling" for the reader.

We've added a paragraph to detail the antiphonal calling behavior in the method section (line 152-160) and clarified in the result section (line 241-242).

3. Be more specific in defining low and high frequency noise --- does this mimic an interfering call? If not, then what? What specific environmental noises are also being simulated. How accurate are the simulated sounds in reproducing interfering calls and true environmental noises?

Previous studies have described the marmoset habitat noise, which included both low-frequency and high-frequency sounds (Morrill et al., 2013). We have added sentences in the result section to explain that we used the two interfering sounds in our experiments to mimic the noises in the natural environment of marmosets (line 225-228, 235-237). Although these interfering sounds do

not fully capture the spectral profile of the habitat noise, they contain the spectral energy that spans the frequency range of the habitat noise. They were designed to model the acoustic contexts in the natural habitat that can be used to test our hypothesis in this study.

4. The perturbation signal occurs "after the onset of a phee call". Is it overlapping the call and to what extent? Would there be a different effect if not overlapping?

The perturbation signal partially overlapped the phee calls. It started with a delay after vocal onset so it mainly overlapped with the latter half of the phees. If they did not overlap with the ongoing calls, marmosets may choose a strategy to shift the time of vocal production to avoid being masked by the perturbation signals (Roy et al., 2011). We have added explanations of this in the result section (line 251-256).

5. Please describe how the virtual calls and the perturbations are actually implemented with the Marmoset calls in real time. The authors say nothing about this!

Our apology! We have added detailed descriptions in the method section (line 147-175).

6. The statistical analysis appears careful but needs further clarity.

We've added additional text to report detailed statistical analysis in the result section (line 327-329, 380-381).

7. In a real dialogue, would a receiving marmoset not get confused if the caller is anticipating a shift not present and thus would the learning effect stop soon?

The reviewer raised an interesting question here. We cannot address this question by the current study. Our experiment focused on the vocal production of individual marmosets but did not explicitly test whether one marmoset would track the vocal features produced by the other marmoset. Our playback calls for the "virtual conspecific" were selected from a group of pre-recorded phee calls from other marmosets and were presented in random order.

8. In the discussion, authors should tone down the generalizations based on the limited experimental setup.

We have revised the discussion to tone down generalizations based on the reported data. We have also added the following sentence at the end of the first paragraph of the discussion section to indicate the limitation of the present study (line 418-420). In addition, in response to Reviewer-1's critiques, we have carefully re-worded a number of statements throughout the manuscript to make sure we do not overstate the findings of this study (please see our responses to Reviewer-1 above).

“A limitation of the present study is the limited experimental parameters tested due to the challenging nature of these experiments. Future studies shall evaluate the above conclusions in a wider range of acoustic contexts.”

9. Figures 1,2 and parts of 3,4 are very confusing in terms of timing of the various signal components. Please clarify.

We have added clarifications in the figure legend.